# Enhanced echocardiographic assessment of intracardiac flow in congenital heart disease

**Brett A. Meyers**[1], **Jiacheng Zhang**[1], **Jonathan Nyce**[2], **Yue-Hin Loke**[2], **Pavlos P. Vlachos**[1] *

1 School of Mechanical Engineering, Purdue University, West Lafayette, IN, United States of America,
2 Division of Cardiology, Children's National Hospital, Washington, DC, United States of America

* pvlachos@purdue.edu

## Abstract

### Background

4D flow magnetic resonance imaging (4D flow MRI) can assess and measure the complex flow patterns of the right ventricle (RV) in congenital heart diseases, but its limited availability makes the broad application of intracardiac flow assessment challenging. Color Doppler imaging velocity reconstruction from conventional echocardiography is an emerging alternative, but its validity against 4D flow MRI needs to be established.

### Objective

To compare intracardiac flow parameters measured by color Doppler velocity reconstruction (DoVeR) against parameters measured from 4D flow MRI.

### Methods

We analyzed 20 subjects, including 7 normal RVs and 13 abnormal RVs (10 with repaired Tetralogy of Fallot, and 3 with atrial-level shunts). Intracardiac flow parameters such as relative pressure difference, vortex strength, total kinetic energy, and viscous energy loss were quantified using DoVeR and 4D flow MRI. The agreement between the two methods was determined by comparing the spatial fields and quantifying the cross-correlation and normalized difference between time-series measurements.

### Results

The hemodynamic parameters obtained from DoVeR and 4D flow MRI showed similar flow characteristics and spatial distributions. The time evolutions of the parameters were also in good agreement between the two methods. The median correlation coefficient between the time-series of any parameter was between 0.87 and 0.92, and the median L2-norm deviation was between 10% to 14%.

### Conclusions

Our study shows that DoVeR is a reliable alternative to 4D flow MRI for quantifying intracardiac hemodynamic parameters in the RV.

**Data Availability Statement:** All relevant data are within the paper and its Supporting Information files.

**Funding:** PPV received support for this project by Grant Number 1R21HD109490 from the National

Institutes of Health, National Institute of Child Health & Human Development. The content is solely the responsibility of the authors and does not necessarily represent the official views of the National Institutes of Health (https://www.nichd.nih.gov). Furthermore, NIH had no role is study design, data collection and analysis, decision to publish, or preparation of the manuscript. No industry partnerships collaborated on or funded this work.

**Competing interests:** I have read the journal's policy and the authors of this manuscript have the following commpeting interests: BAM and PPV have intellectual property filings for each analysis algorithm in the analysis method used in this manuscript. BAM and PPV are involved with Cordian Technologies, a start-up founded by PPV, which licenses the technologies used in this work. YHL receives partial salary support from NHLBI (R21-HL156045-01). These disclosures do not alter our adherence to PLOS One policies on sharing data and materials.

## Introduction

Congenital heart disease (CHD), particularly those involving the complex right ventricle (RV), is characterized by structural and blood flow abnormalities necessitating lifelong non-invasive imaging surveillance. For example, patients with repaired Tetralogy of Fallot (TOF) may experience RV failure due to deleterious effects of chronic pulmonary insufficiency [1, 2]. Transthoracic echocardiography (echo) serves as the "frontline" modality for assessing RV structure and function; however, its ability to characterize pathological intracardiac flow patterns is limited to qualitative observations. Cardiac magnetic resonance (CMR) imaging offers enhanced visualization and measurements, but conventional parameters such as end-diastolic/end-systolic volumes and ejection fraction do not adequately capture the spectrum of flow abnormalities in CHD patients [3].

Current imaging research have developed RV measurements detectable by both echo and CMR such as intracardiac flow dynamics [4, 5]. Various methods exist for measuring blood flow velocity, including color Doppler echo, blood speckle tracking, and CMR-based phase-contrast imaging. Among these, four-dimensional flow (4D flow), an advanced CMR technique, stands out as it enables detection of all three velocity component vector fields within a three-dimensional (3D) space. However, the practical use of 4D flow is hindered by availability, considerable time and resource requirements, and the need for sedation in pediatric patients. These factors present significant challenges for employing 4D flow MRI in serial monitoring of blood flow in CHD.

Color Doppler echocardiography has been proposed as a viable alternative to 4D flow by capturing highly resolved axial component of blood velocity and then reconstructing the lateral component inside the intraventricular volume of interest. Vector flow mapping (VFM) is the most commonly used reconstruction tool [6], but its agreement with 4D flow MRI has not been demonstrated quantitatively. Blood speckle tracking (BST) is another technique that has not yet been validated clinically [7]. Both VFM and BST are vendor-specific, relying on specific equipment and echo acquisitions. To that end, Doppler velocity reconstruction (DoVeR) is a newly developed reconstruction method enabling analysis of non-vendor-specific color Doppler imaging [8].

This study aimed to validate advanced intracardiac flow parameters (relative pressure, vorticity, kinetic energy, and flow energy loss) measured from echo-based DoVeR method against 4D flow MRI. We quantitatively and qualitatively compared DoVeR from echo imaging acquired in the apical four-chamber (A4C) plane against 4D flow MRI projected onto the same plane. We hypothesized that RV measurements of DoVeR would be in good agreement with 4D flow measurements in both CHD and controls.

## Methods

### Study population

In a study approved by the Institutional Review Board (IRB) of both Purdue University (IRB-2021-64) and Children's National Hospital (Pro00010769), echo imaging and 4D flow MRI from CHD and control subjects between January 1st, 2018, and October 3rd, 2020, was retrospectively obtained. All studies were performed within the Children's National Hospital system. Patient data for use in this study was identified and collected from October 2020 to June 2021. Included subjects required diagnostic-level echo visualization of the RV cavity from the A4C plane with an accompanying CMR study within 1 year of the echo. CHD subjects included those with repaired Tetralogy of Fallot (rTOF), right ventricle dilation (RVD) from sinus venous defects with partial anomalous pulmonary venous return (Qp:Qs > 1.2:1), and

structurally normal hearts (control). The control subjects were obtained for a separate clinical indication and were found to have normal RV size, function, and pulmonary-to-systemic flow ratio (Qp:Qs < 1.2:1). Studies with inadequate echo imaging quality and significant stent/sternal artifacts precluding CMR 4D flow analysis were excluded. Because this study was retrospectively performed, informed consent was not required by either IRB protocols. All records shared with Purdue University were anonymized to remove personal information identifiers.

## Echocardiogram acquisition

Echo imaging was all performed on a Phillips EPIQ (Philips Healthcare, Andover, MA, USA) using either a 5MHz or 8/9MHz probe depending on patient habitus. Imaging was performed following lab standard imaging protocols for RV assessment. These include adequate visualization of the RV free wall motion for B-mode image acquisitions, and sector width set to the appropriate size to cover the RV while also maintaining a frame rate > 20 Hz for both B-mode and color Doppler acquisitions. A routine A4C view of the RV with color Doppler imaging for at least two cardiac cycles was anonymized and collected. Appropriate echo studies had color Doppler acquisition box over a majority of the RV chamber (particularly with an A4C alignment that adequately resolves both the tricuspid inflow and RV apex), adequate temporal resolution (at least 10 frames per beat) adequate visualization of the RV free wall/septum, and appropriate color scaling to minimize aliasing of the flow.

## CMR acquisition

All CMR imaging was obtained on a Siemens 1.5-T scanner (Siemens Healthineers, Erlangen, German), including long-axis and short-axis cine imaging along with contrast-enhanced 4D flow. 4D flow was performed using WIP785 which is a retrospective, respiratory-gated sequence that utilizes 3D Cartesian sampling with flow encoding, GRAPPA acceleration. For 4D flow, the field of view (FOV) was $280–480 \times 140–230$ mm, with a matrix of $160 \times 77$. The time to echo was 2.19 ms, and the repetition time was 37.9–59.4 ms, dependent on the number of segments per RR interval. The flip angle was 15˚. The slice thickness depended on the patient's size, either 1.8 mm for body surface area (BSA) < 1.5 $m^2$ or 2.8 mm for BSA > 1.5 $m^2$. The cardiac cycle was reconstructed into 20–30 phases. Velocity encoding was set between 2 m/s and 2.5 m/s.

## Doppler velocity reconstruction

DoVeR analysis was performed offline on the color Doppler acquisitions to obtain the blood-flow velocity vector field within the RV. A suite of semi-automated algorithms developed in-house with MATLAB (The Mathworks, Natick, Massachusetts) was used to initialize each study by selecting three feature points for three frames. The RV is segmented from each frame using a machine vision-based algorithm to identify the endocardial boundary [9]. DoVeR reconstructs the two-component and two-dimensional velocity vector field using the velocity measurements and initial conditions for each color Doppler frame [8]. A DoVeR reconstruction for a single dataset takes approximately 7 minutes to run, which covers the steps outlined above.

The DoVeR-derived velocity vector fields were further processed to construct the velocity vector fields corresponding to a single heartbeat sequence. The velocity fields for each scan were decomposed to a lower-order representation [10], then beats were phase-averaged and resampled to match the temporal resolution of the corresponding 4D Flow MRI recording.

## 4D flow post-processing

Commercially available software (iTFlow; Cardio Flow Design Inc., Tokyo, Japan) [11] was used to segment the 4D Flow MRI velocity fields. Spurious velocity vectors were detected using the Universal Outlier Detection method [12]. The outlier velocity values were replaced with the median of the neighborhood of local non-spurious velocity values.

## Quantitative analysis

The velocity vector fields obtained from 4D flow MRI and echo-derived DoVeR was then used to calculate intracardiac flow parameters. These parameters include inflow velocity based on velocity magnitude ($|V|$), vortex strength (VS), viscous energy loss (VEL), total kinetic energy (KET), and intraventricular pressure difference (IVPD) [13]. The inflow velocity was measured from the tricuspid valve to 1 cm into the RV as the 97.5% quantile for each frame. The VS is based on vorticity ($\overrightarrow{\omega}$), the potential for fluid rotation, integrated over the ventricle area. Vorticity is computed taking the curl of the velocity field $\overrightarrow{v}$,

$$\overrightarrow{\omega} = \nabla \times \overrightarrow{v} \tag{1}$$

The VS is a surrogate for the total fluid rotation for all vortex cores in the field and is computed by integrating vorticity over the ventricle area,

$$VS = \int_A |\overrightarrow{\omega}| \cdot dS \tag{2}$$

Viscous energy loss (VEL) describes the amount of fluid energy dissipated in the flow, which can include fluid rotation, stagnation, acceleration, or deceleration. This quantity is computed as the integral of the summed spatial velocity gradients for each pixel or voxel over the RV volume,

$$VEL = \int_V \frac{1}{2} \mu \sum_{ij} \left( \frac{\partial v_i}{\partial x_j} + \frac{\partial v_j}{\partial x_i} \right)^2 dv \tag{3}$$

Here, $\mu$ is the blood viscosity and *i,j* are indices representing each spatial component of the velocity vector field. Eq 3 assumes the velocity field is divergence-free, which is inherently satisfied by DoVeR and enforced in the 4D flow MRI post-processing.

Kinetic energy (KET) is the amount of energy due to flow or motion. This quantity is computed as the integral of the summed squared $\overrightarrow{v}$ for each pixel or voxel over the RV volume,

$$KET = \int_V \frac{1}{2} \rho \sum_i v_i^2 \, dv \tag{4}$$

The intraventricular pressure difference (IVPD) is computed from the relative pressure fields reconstructed from the velocity vector fields in the RV. The pressure gradient in the RV was first evaluated from the velocity vector fields based on the Navier Stokes Equation (NSE) as,

$$\frac{\partial P}{\partial x_i} = -\rho \left( \frac{\partial u_i}{\partial t} + u_j \frac{\partial u_i}{\partial x_j} \right) + \mu \frac{\partial^2 u_i}{\partial x_j \partial x_j} \tag{5}$$

Here $\mu$ is the density of blood. The relative pressure field was then obtained by integrating the pressure-gradient with weighted least-squares [13] as,

$$p = (\| W(Gp - \nabla p) \|) \tag{6}$$

where $p$ is the column vector consisting of the reconstructed relative pressure, $G$ represents the discrete gradient operator constructed using the second order central (SOC) difference scheme, $\Delta p$ is the column vector of the evaluated pressure-gradient values, $\|\cdot\|$ represents the L2 norm, and $W$ is the weight matrix generated based on the pressure-gradient errors. The IVPD is calculated as the difference between pressure at the tricuspid valve ($P_{TV}$) and the RV apex ($P_{Apex}$) such that,

$$IVPD = \Delta P = P_{TV} - P_{Apex}. \qquad \text{Eq7}$$

## Statistical analysis

Statistical analysis was performed using MATLAB statistical analysis toolbox on timeseries measurements of the volume-averaged flow dynamics parameters. Additionally, we included The correlation of timeseries measurements between echo-derived DoVeR and 4D flow MRI was assessed using Pearson's correlation coefficient (R), and good agreement was defined as R greater than 0.7. The difference in magnitude between echo-derived DoVeR and 4D flow MRI timeseries measurements was quantified by root mean square difference or L2-norm difference. Good agreement was defined as L2-norm less than 25%. Population demographics were analyzed using a one-way analysis of variance, and quantities were considered statistically significant if $p < 0.050$.

# Results

## Study population demographics

Our study included 12 rTOF patients, 5 RVD patients, and 9 controls, as shown in Table 1. Any possible selection bias cannot impact the results, as this was a cross-platform validation study for each patient. Reported RV measurements are collected from CMR imaging. RV Ejection Fraction (EF) was lower in rTOF patients than in RVD patients and controls ($49 \pm 6\%$ vs. $63 \pm 6\%$ and $60 \pm 4\%$, $p = 0.001$). A total of 10 rTOF patients, 3 RVD patients, and 7 controls successfully underwent reconstruction. For 6 cases (23%), either poor color Doppler acquisition settings or poor CMR alignment, where the 4D flow datasets did not have enough slices to cover the RV apex fully, prevented further analysis. The average color Doppler recording frame rate was 20 Hz, the penetration depth did not exceed 15 cm into the chest, and the sweep region covered the RV and right atrium.

**Table 1. Population demographics.** BSA: body surface area, RVEDVi: right ventricular indexed end-diastolic volume, RVESVi: right ventricular indexed end-systolic volume, EF: ejection fraction, PR: pulmonary valve regurgitant fraction, Qp:Qs: pulmonary to systemic blood flow ratio.

| | rTOF (n = 12) | RVD (n = 5) | Control (n = 9) | p-value |
|---|---|---|---|---|
| Age (years) | 15.9 ± 8.7 | 9.2 ± 5.8 | 11.4 ± 6.4 | 0.328 |
| Gender (F) | 12 (8) | 5 (4) | 9 (5) | 0.222 |
| BSA (m$^2$) | 1.4 ± 0.4 | 1.1 ± 0.6 | 1.2 ± 0.5 | 0.642 |
| RVEDVi (mL/ m$^2$) | 143.0 ± 49.8 | 182.9 ± 32.0 | 82.0 ± 17.1 | 0.498 |
| RVESVi (mL/ m$^2$) | 74.6 ± 32.9 | 66.7 ± 3.7 | 33.3 ± 8.0 | 0.066 |
| EF (%) | 49.2 ± 6.2 | 62.8 ± 6.1 | 59.5 ± 3.9 | **0.001** |
| PR (%) | 31.6 ± 17.7 | - | - | - |
| Qp:Qs | 1.01 ± 0.07 | 2.68 ± 0.30 | 1.03 ± 0.10 | **< 0.001** |

BSA: body surface area, RVEDVi: right ventricular indexed end-diastolic volume, RVESVi: right ventricular indexed end-systolic volume, EF: ejection fraction, PR: pulmonary valve regurgitant fraction, Qp:Qs: pulmonary to systemic blood flow ratio

## Similarity of DoVeR and 4D flow MRI flow dynamics

This study focuses on the A4C plane and visualizes the peak diastole phase of the cardiac cycle. During this phase, a high-velocity jet forms as the tricuspid valve opens, pulling blood from the right atrium into the RV and causing a vortex ring to develop in the basal region that promotes filling.

Fig 1 presents the flow fields within the RV at peak diastole using a normal patient as the representative case. The stream tracers from (1) DoVeR and (2) 4D flow MRI represent the flow fields, with the color scale indicating the local quantity of (a) velocity, (b) pressure, (c) vorticity, (d) kinetic energy, and (e) energy loss. Timeseries measurements quantified over the RV volume (3) are also provided. To enable visual and qualitative comparison with DoVeR, the 4D flow MRI measurements are projected from the volume onto the A4C plane. To project the 4D flow MRI measurements, the intracardiac flow was first isolated from a 3D segmentation, and then velocity measurements were interpolated linearly from the preserved 4D flow data onto a 2D plane parallel to the A4C view.

The stream tracers show a donut-shaped vortex ring with prominent vortex core features near the basal free wall and the septal mid-wall. The 4D flow stream traces capture the free wall vortex, while the DoVeR stream traces capture both the mid-wall and the free wall vortices. The magnitude of the filling jet is similar between modalities, but the penetration depth and stagnation occur at different locations, as can be seen in Fig 1A-1 and 1A-2.

Relative pressure fields (Fig 1B) show low pressure at the vortex core centers and high pressure toward the apex region where flow stagnates. The tricuspid valve-to-apex pressure difference is associated with pressure recovery which assists in proper valve closure. Elevated vorticity (Fig 1C) occurs at the vortex centers for both methods, along with high kinetic energy within the high-velocity jet (Fig 1D). Energy loss appears similar in magnitude between modalities (Fig 1E), but DoVeR exhibits greater losses near the valve and leading edge of the jet, while 4D flow shows greater losses along the edge of the vortex core due to the inflow jet.

Fig 1A-3 through 1E-3 show the timeseries measurements for the inflow velocity, intraventricular pressure difference, vorticity, kinetic energy, and energy loss for an anatomically normal RV. Timeseries measurements for vorticity, kinetic energy, and energy loss are indexed by RV area (color Doppler) or volume (4D flow MRI). Generally, the timeseries measurements for both modalities exhibit similar characteristics, with differences in magnitude. During early diastole, a positive pressure difference is generated between the tricuspid annulus and apex, which becomes negative as the RV filling rate slows between 0.5 T and 0.7 T. The highest measurements for vortex strength, energy loss, and kinetic energy are observed around 0.5 T, corresponding to the peak inflow jet during diastole.

## Agreement of timeseries measurements

For the representative case shown in Fig 1, the timeseries measurements appear in good agreement between modalities. The correlation coefficient is highest (0.98) for pressure difference measurements and lowest (0.82) for vortex strength measurements. Additionally, the L2-norm error was smallest (1.2%) for the pressure difference measurements and largest (14.7%) for the viscous energy loss measurement. This case demonstrates that DoVeR and 4D flow MRI measurements can be highly correlated.

Fig 2 displays the statistical distributions of the R and L-2 norms for the flow dynamics parameters for all subjects. Table 2 provides the quartiles from the distribution for each parameter. The median R for each parameter ranges from 0.80 to 0.92, with the lowest quartile R at 0.74. The median L2-norm for each parameter ranges from 8.4% to 13.2%, with the highest

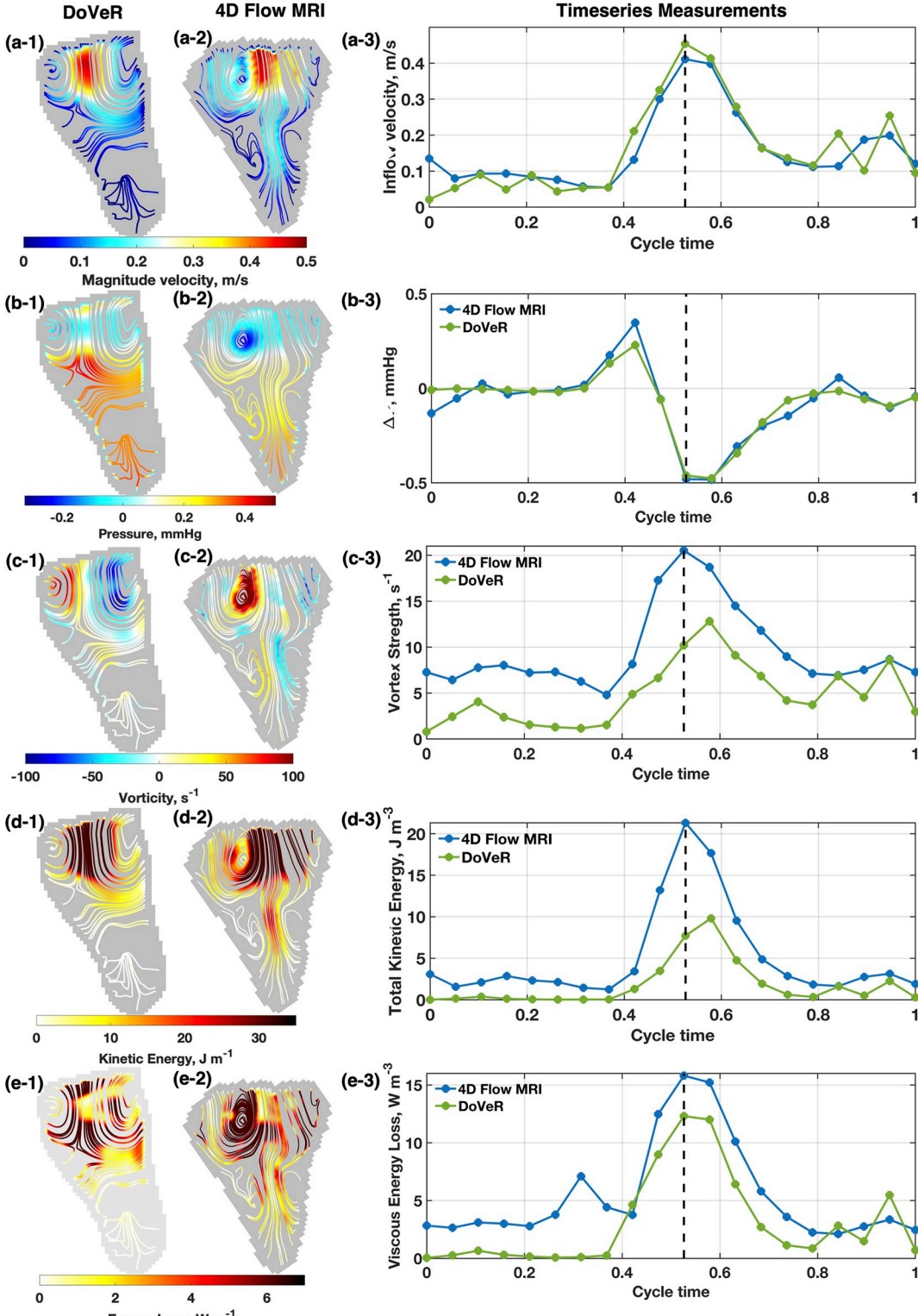

**Fig 1. Comparison of DoVeR and 4D flow MRI during early filling.** Streamlines are used to present the flow field for both (1) DoVeR and (2) 4D flow MRI modalities. (3) Volume-averaged timeseries measurements are presented to show agreement between methods across the cardiac cycle. Flow parameters shown include (a) velocity, (b) relative pressure, (c) vorticity, (d) kinetic energy, and (e) energy loss. During early diastole inflow a donut shaped vortex ring forms near the tricuspid valve around the inflow jet, which is captured by the streamlines provided for both modalities. Flow fields and timeseries measurements are characteristically similar, however there are magnitude differences for some cases.

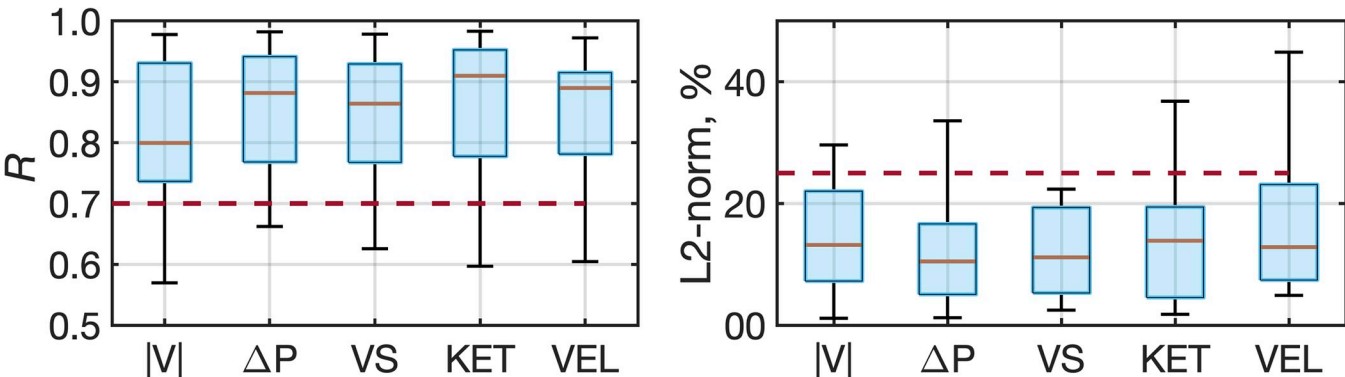

**Fig 2. Distribution of correlation coefficients and L2-norm across modalities.** The dotted red line corresponds to the threshold for good agreement for the correlation coefficient (R = 0.7) and normalized L2-norm (25%). |V|: velocity magnitude; ΔP: tricuspid valve to apex pressure difference; VS: vortex strength; KET: total kinetic energy; VEL: viscous energy loss.

quartile at 22.0%. Most timeseries between echo-derived DoVeR and 4D flow MRI exhibit favorable agreement for both R and L2-norm.

## Flow pattern differences across conditions

Fig 3 shows flow fields for different cardiac health conditions during the early diastole phase of the cardiac cycle. The subjects were selected based on age (~10 years old) and BSA (~1.0 $m^2$). By examining these three demonstration cases, one can observe the similarities and differences between imaging modalities and conditions. Additional peak and average parameter statistics for all conditions are reported within Table 1 in S1 Table. We direct interested readers here to further understand the differences between observed quantities in this study.

The normal and RVD patients have similar flow field appearances, while the rTOF patient shows more notable variations. In both the normal and RVD hearts, the inflow jet induces formation of vortices along the free wall and septal mid-wall. The relative pressure fields show positive pressure toward the apex and negative pressure at the tricuspid valve, and result in similar pressure differences ($\Delta P \approx -0.4 mmHg$). The inflow jet concentrates energy loss, with viscous energy loss being similar across image modalities for the normal case ($VEL \approx -0.4 mmHg$), while the DoVeR measurement is slightly elevated compared to the 4D flow MRI for the RVD case ($VEL_{DoVeR} \approx 40\ W/m^3$, $VEL_{4D\ flow\ MRI} \approx 25\ W/m^3$).

In the rTOF patient, a vortex formation along the free wall vortex is present, but it is smaller compared to the other conditions, and the 4D flow MRI fails to capture any vortex formation within the A4C plane. Additionally, the flow patterns from the 4D flow MRI appear more disordered. There is less agreement between the DoVeR and 4D flow MRI measurements for both the pressure difference ($\Delta P_{DoVeR} \approx -1 mmHg$, $\Delta P_{4D\ flow\ MRI} \approx -0.5 mmHg$) and viscous energy loss ($VEL_{DoVeR} \approx 20\ W/m^3$, $VEL_{4D\ flow\ MRI} \approx 40\ W/m^3$).

## Disscussion

This study aimed to compare the efficacy of an in-house developed and vendor-independent echo Color Doppler velocity reconstruction method with 4D flow MRI in quantifying intracardiac flow fields of the RV in congenital heart disease patients. The study found good agreement in variables such as VS, VEL, KET, and IVPD between DoVeR and 4D flow MRI. This study is the first to validate the inter-modality agreement of novel flow dynamic parameters between a color Doppler velocity reconstruction method and 4D flow MRI.

**Table 2. Statitiscal analysis of correlation coefficient and L2-norm differences.** The quartiles of the distributions of R and relative L2-norm difference between the time-series measurements of each hemodynamic parameter from DoVeR and 4D flow MRI. |V|: velocity magnitude; ΔP: tricuspid valve to apex pressure difference; VS: vortex strength; KET: total kinetic energy; VEL: viscous energy loss.

| Hemodynamics | Correlation coefficient (R) | | | | | Relative L2-norm difference, % | | | | |
|---|---|---|---|---|---|---|---|---|---|---|
| | \|V\| | $\Delta P$ | VS | KET | VEL | \|V\| | $\Delta P$ | VS | KET | VEL |
| 25th | 0.74 | 0.83 | 0.77 | 0.81 | 0.81 | 7.3 | 5.0 | 5.1 | 4.4 | 7.1 |
| 50th | 0.80 | 0.88 | 0.87 | 0.92 | 0.89 | 13.2 | 8.4 | 10.8 | 13.1 | 12.7 |
| 75th | 0.93 | 0.95 | 0.93 | 0.96 | 0.92 | 22.0 | 16.1 | 17.4 | 18.4 | 21.5 |

Accurate, in-vivo measurements of intracardiac flow is crucial to the investigation of cardiac biomechanics. Previous studies using 4D flow MRI have identified abnormalities in the diastolic vortex, KET and VEL in certain patient populations [5, 14–17]. At the same time, overreliance on this method may lead to disparities in healthcare management [18, 19] and is a barrier to broader clinical adoption of these measurements. Conversely, color Doppler echo is widely available, but there is a low level of agreement between this modality and CMR [20].

This study demonstrates that DoVeR and 4D flow provide similar flow characteristics and agreement in the RV, as shown in Figs 1 and 3. This finding is significant as there are uncertainties regarding the ability for a 2D planar field to capture complex 3D characteristics of RV shape and intracardiac flow fields observed by 4D flow MRI. During normal diastole, the tricuspid inflow generates a vortex ring that propagates flow towards the outflow tract with minimal contribution to the apex. However, flow patterns can be disrupted by factors such as pulmonary insufficiency in rTOF patients, as observed in the rTOF subject in Fig 3 whose disparities may be attributed to the substantial pulmonary regurgitation ($PR\approx35\%$) experienced. Despite this potential limitation, the R and L2-norms of the instantaneous differences between timeseries measurements demonstrates good agreement between the two modalities, indicating that DoVeR could potentially offer comparable spatial distribution and temporal evolution of flow dynamics parameters that are comparable to 4D flow MRI. Furthermore, as DoVeR relies on "frontline" echo, it has an advantage in serial investigation of intracardiac flow parameters for CHD patients, which is not practical using 4D flow MRI.

Flow visualization like DoVeR and 4D flow MRI facilitate a deeper understanding of the flow fields within the heart, shedding light on alterations under different disease conditions, as illustrated in Fig 3. In a larger cohort of rTOF patients, we observed that the RV had significant pulmonary regurgitation had more disturbed flow and a less prominent tricuspid vortex ring formation compared to both normal controls [21]. This phenomenon appears to be associated with exercise intolerance in rTOF patients [5]. Thus, future studies employing DoVeR could elucidate the extent of these deviations and identify potential parameters for characterizing patient health and potentially predicting outcomes, such as the optimal timing for pulmonary valve replacement.

Other echo based VFM studies have attempted to assess the agreement with 4D flow MRI with limited results. One study noted that kinetic energy distributions in velocity vector maps using VFM were similar to those obtained with 4D flow MRI [22]. Another study compared a simple algorithm for identifying vortex structures from color Doppler echocardiography against VFM and 4D flow MRI modalities but did not directly compare the latter techniques [23]. In a recent study, agreement of blood flow topology measurements was studied between VFM and 4D flow MRI measurements; however the sample size was small with modest agreement and did not assess the performance of VFM in detecting flow alterations [24]. These may be the result of inherent limitations to the VFM formulation–as VFM purely relies on conservation of flow, there are inherently higher sources of error when conservation of momentum

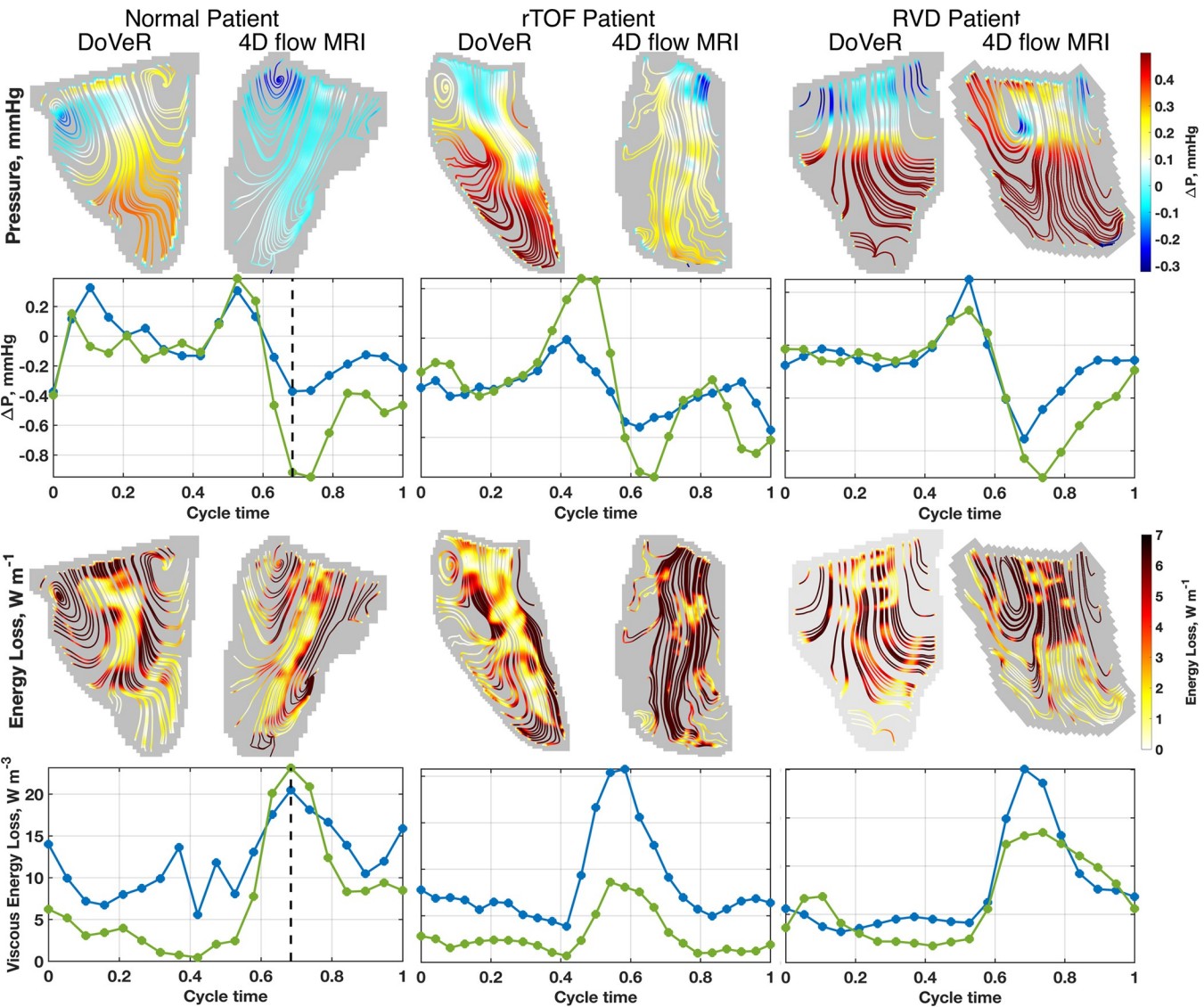

**Fig 3. Early filling patterns for different cardiovascular health conditions.** Results shown for DoVeR and 4D flow MRI. Flow fields are shown for relative pressure and energy loss intracardiac flow parameters. (Left) The 4D flow MRI relative pressure appears lower than the DoVeR measurement, however the valve-to-apex differences are similar; the energy loss appears concentrated along the jet inflow for both measurement modalities. (Middle) Relative pressure appears in better agreement between measurements, while the 4D flow MRI energy loss appears considerably elevated compared to the DoVeR method. (Right) Both relative pressure and 4D flow MRI measurements appear in good agreement.

is not accounted for, thus affecting the quality of the parameters. BST, a correlation-based displacement estimation method, provides a more physiologically consistent measurement of intracardiac blood flood; this method has been validated against flow phantoms and computational models [7, 25] but still lacks clinical validation. Furthermore, BST is mostly limited to near-field assessments [7, 26], though application with adult patients has been demonstrated [27]. These factors, coupled with the fact that BST is only provided by a single vendor, contribute toward lower availability in clinical settings. By relying on conventional imaging and incorporating the appropriate fluid dynamic assumptions, we hope that DoVeR can solve the significant technological gaps in the clinical investigation of intracardiac flow to discern disease course in CHD.

Despite the agreement between DoVeR and 4D flow, discrepancies between the two modalities persist, which can be attributed to multiple factors. Misalignment between the color Doppler imaging plane and the A4C plane for projecting 4D flow data can introduce additional differences when comparing the two modalities. Calculations of hemodynamic parameters are affected by the spatial resolution of the velocity data, and the different spatial resolutions of DoVeR and 4D flow MRI can lead to variance in hemodynamic parameters. Furthermore, as the DoVeR assessment focuses on the A4C plane, other sources of inflow contributing to VEL and KET are not captured, particularly pulmonary insufficiency. This likely contributes to the finding shown in Fig 1, where L2-norm between modalities is higher for VEL and KET. Beat-to-beat variability, intra- and inter-operator variability, and intra-modality variability can also affect the inter-modality comparison. A partial contribution to the variability stems from the 1 year time difference between the echocardiogram and CMR recording sessions. This factor may not be critical for normal and stable patients but may lead to greater discrepancy in patients with worsening valve insufficiency in the rTOF population. Optimizing the 4D flow projection to the A4C plane, sufficient sampling of color Doppler scans to build an ensemble like MRI, and resampling DoVeR to match the MRI resolution can help reduce discrepancies.

Our study has limitations that need acknowledgment. We had 26 subjects in our cohort, with varying numbers in each condition subgroup. Although our study aim is to compare DoVeR and 4D flow MRI agreement, differences in performance across conditions might exist. Additionally, 8 subjects (23% of the cohort) were excluded, potentially further weakening our measurements' significance. While we explored differences in R and L2-norm between conditions, no statistically significant findings were observed, possibly due to the smaller subgroup sizes. This could impact the power and significance of agreement between subgroups and underscores the need for a larger sample in future studies. Because this study used retrospective acquisitions of clinical data, no specific protocols to maximize DoVeR capabilities were utilized. This was an intentional aspect of our study design to assess the robustness of DoVeR for use with retrospective clinical imaging. Future studies will work to establish proper protocols to maximize DoVeR for use in RV flow analysis.

The DoVeR method similarly has a few limitations. Its current formulation is solely in 2D, potentially missing key aspects of highly 3D flows, like those in the RV, impacting the capture of relevant flow features. Our future focus includes expanding this method to 3D for reconstructing flows observed in 3D color Doppler recordings of the RV. DoVeR reconstruction can fail to provide a reliable reconstruction if the RV free wall is absent, improper tracking of the tricuspid annular plan occurs, or insufficient color signal is present during diastolic inflow. In these cases, either an inconsistent segmentation is rendered, which leads to an erroneous velocity field output, or the underlying flow physics cannot be properly satisfied. Addressing these limitations is an ongoing focus of our research efforts.

This study confirms the utility of color Doppler velocity reconstruction for assessing intra-cardiac flow in the RV of pediatric patients with CHD. The flow characteristics and spatial fields of hemodynamic parameters obtained from DoVeR are comparable to those from 4D flow MRI, with high correlation and strong agreement. As such, DoVeR offers a viable alternative for quantifying cardiac flow conditions, which has the potential to enhance investigations into cardiac function in CHD.

## Supporting information

**S1 File. Data used to generate Fig 2 and Table in S1 Table.** A file containing all measurements used to generate results presented in this study.
(XLSX)

**S1 Table. DoVeR and 4D Flow measurements obtained from analysis.** A table of reported means, standard deviations, and p-values for the flow measurement quantities obtained for each condition across each imaging modality.
(DOCX)

**S1 Text. Abbreviations.** A summarizing table of all abbreviations not commonly used in practice.
(DOCX)

## Author Contributions

**Conceptualization:** Yue-Hin Loke, Pavlos P. Vlachos.

**Data curation:** Jonathan Nyce, Yue-Hin Loke.

**Formal analysis:** Brett A. Meyers, Jiacheng Zhang, Pavlos P. Vlachos.

**Funding acquisition:** Pavlos P. Vlachos.

**Investigation:** Brett A. Meyers, Jonathan Nyce, Yue-Hin Loke, Pavlos P. Vlachos.

**Methodology:** Brett A. Meyers, Jiacheng Zhang, Pavlos P. Vlachos.

**Project administration:** Yue-Hin Loke, Pavlos P. Vlachos.

**Resources:** Yue-Hin Loke, Pavlos P. Vlachos.

**Software:** Brett A. Meyers, Jiacheng Zhang, Pavlos P. Vlachos.

**Supervision:** Yue-Hin Loke, Pavlos P. Vlachos.

**Visualization:** Brett A. Meyers, Jiacheng Zhang.

**Writing – original draft:** Brett A. Meyers.

**Writing – review & editing:** Brett A. Meyers, Jiacheng Zhang, Jonathan Nyce, Yue-Hin Loke, Pavlos P. Vlachos.

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
