## [Decision Letter · Decision Letter 0]

7 Dec 2023

PONE-D-23-30513Enhanced echocardiographic assessment of intracardiac flow in congenital heart diseasePLOS ONE

Dear Dr. Vlachos,

Thank you for submitting your manuscript to PLOS ONE. After careful consideration, we feel that it has merit but does not fully meet PLOS ONE’s publication criteria as it currently stands. Therefore, we invite you to submit a revised version of the manuscript that addresses the points raised during the review process.

We look forward to receiving your revised manuscript.

Kind regards,

Hany Mahmoud Abo-Haded, MD

Academic Editor

PLOS ONE

“PPV received support for this project by Grant Number 1R21HD109490 from the National Institutes of Health, National Institute of Child Health & Human Development. The content is solely the responsibility of the authors and does not necessarily represent the official views of the National Institutes of Health (https://www.nichd.nih.gov). No industry partnerships collaborated on or funded this work.”

“I have read the journal's policy and the authors of this manuscript have the following

commpeting interests:

BAM and PPV have intellectual property filings for each analysis algorithm in the

analysis method used in this manuscript.

BAM and PPV are involved with Cordian Technologies, a start-up founded by PPV,

which licenses the technologies used in this work.

YHL receives partial salary support from NHLBI (R21-HL156045-01).”

Additional Editor Comments:

Please reply to all the reviewers' concerns

Reviewers' comments:

Reviewer's Responses to Questions

**Comments to the Author**

1. Is the manuscript technically sound, and do the data support the conclusions?

Reviewer #1: Yes

Reviewer #2: Yes

Reviewer #3: Yes

2. Has the statistical analysis been performed appropriately and rigorously? 

Reviewer #1: Yes

Reviewer #2: Yes

Reviewer #3: Yes

3. Have the authors made all data underlying the findings in their manuscript fully available?

Reviewer #1: Yes

Reviewer #2: Yes

Reviewer #3: No

4. Is the manuscript presented in an intelligible fashion and written in standard English?

Reviewer #1: Yes

Reviewer #2: Yes

Reviewer #3: Yes

5. Review Comments to the Author

Reviewer #1: 1. Were the echo and CMR imaging all done at one institution, or were there patients from both centers?

2. Inclusion criteria are subjects with “diagnostic-level echo visualization of the RV,” then there are patients later excluded for poor color Doppler alignment. Can you expand on what is needed for color Doppler alignment for adequate analysis with DoVeR? In general, how challenging is it to obtain adequate echo imaging of the RV for DoVeR. I can see this being especially difficult in the adult congenital population where it also may be most useful.

3. What was the reason for MRI on the cohort of normal patients?

4. The text states for Figure 1 (d) as energy loss, and (e) as kinetic energy, whereas the figure states the opposite. Please clarify.

5. Please expand how the 4D flow measurements are projected onto 2D space? Is this for visualization similar to a “maximal intensity projection (MIP)” would be for an angiogram? Is this for visualization only, or for quantification as well? Do the 4D flow measurements account for all of 3D space?

6. I understand that the small numbers of this study make this challenging, though can you discuss differences in the agreement of parameters between DoVeR and 4D flow in the different patient groups. Is there generally more agreement in normal patients vs rTOF patients as DoVeR is missing the RVOT which is captured in 4D flow?

7. Would there be utility in repeating this analysis in other echocardiographic planes to capture RV outflow and pulmonary regurgitation?

Reviewer #2: In this article, the authors aimed to compare the efficacy of echo DoVeR method with 4D flow MRI in quantifying intracardiac flow fields of the RV in some patients with congenital heart disease and controls. In general, the manuscript is well-structured and provides a clear context for your study. I have a few comments/suggestions:

Methods section:

1) Pg. 4, Line 74: The patient recruitment phase concluded in October 2020, extending beyond three years. Were there any additional patient enrolments during this period, and could you provide details regarding the study's timetable throughout? Additionally, given the retrospective nature of the study, it would be beneficial to specify the commencement date for the analysis of echo and MRI images. Clarifying this information will enhance the transparency of the study timeline.

2) Pg 4, Line 77: You enlisted three patients with right ventricular dysfunction (RVD) stemming from "atrial-level shunting." It would be helpful to clarify whether this atrial-level shunting exclusively involved atrial septal defects (ASD) in these patients or if other congenital heart diseases, such as partial anomalous pulmonary venous drainage, were also present. If the RVD was solely attributed to ASD, it might be more precise to indicate RVD in patients with ASD, avoiding the term "atrial-level shunting" to prevent confusion among readers. However, if other pathologies were involved in these three patients, please specify the nature of these conditions.

3) Pg 5, Line 83: “Echocardiogram Acquisition”. Provide a bit more detail on the echo imaging process. For example, mention the echo machine settings used, measures used to improve the temporal resolution and colour Doppler of your images, or any specific protocols followed during acquisition.

4) Pg 5, Line 85: “A routine A4C view of the RV”. I guess you mean A4C focused RV view to better visualize the free RV wall.

5) Pg 6, Line 109: It's mentioned that a single DoVeR reconstruction takes approximately 7 minutes. While this information is useful, you may want to discuss the overall time required for the entire analysis process for each patient, including any preprocessing steps.

6) Pg 6, Line 114: “4D flow post-processing”. You mentioned that the velocity encoding was set between 2 m/s and 2.5 m/s during acquisition. Did you need VENC correction for any of your patients during processing?

Results section:

1) Pg. 9, Line 161: What is meant be poor CMR alignment? Please clarify.

Discussion section:

Discussion of the study's limitations is crucial. Firstly, the limited size of the cohort, with only 13 patients compared to 7 controls, raises concerns. To ensure the robust validation of the echo DoVeR against 4D MRI, a larger sample size is recommended. Secondly, the utilization of single plane 2D echocardiography for reconstructing flow in the complex 3D geometry of the RV is another limitation. Consideration of future endeavours involving coloured 3D echocardiography of the RV, coupled with your semi-automated MATLAB algorithms, could enhance the study. This approach has the potential to reconstruct a 3D colour Doppler velocity, providing a more comprehensive three-dimensional dataset that may better align with data obtained from 4D MRI.

Table 1:

1) I believe that RV measurements (EDVi, ESVi, EF, and PR%) are all obtained from CMR not echocardiography. This should be stated.

2) Is there a particular rationale behind not assessing the QP/QS ratio in patients with rTOF? Some of these patients may have residual shunts across the VSD patch. Even with the presence of severe PR, it's possible to quantify the QP/QS ratio in these individuals using CMR. Could you provide clarification on this matter?

Figures:

1) The quality of the figures is suboptimal and requires enhancement.

Reviewer #3: The authors compared a novel technique for estimating intracardiac flow dynamics from echocardiography against the more established but not widely available 4d flow MRI. The technique and the results are interesting. I have no major comments on the used methodology.

Minor comments:

1- Line 116: used to segment…

2- Line 190: define acronym on first use (CFI)

3- Figure 1: please indicate where the time series is being measured. Is it the mean time series of the whole RV area or taken from a specific point/region?

4- Figure: likewise done with a healthy control in Figure 1, it would be interesting to show the time series from a representative CHD patient with low RV function. Or maybe add the time series to the slices shown in Figure 3.

5-

Results section: consider moving any interpretations of the results to the discussion, e.g. line 227-228.

6-

The values in Table S1 are of great interest. Consider moving these to the main manuscript body and cite the table in the text.

7-

Discuss the limitation of small study participants (especially after excluding 8 cases due to failed reconstruction). Also discuss factors of reconstruction failure as a limitation of the technique.

6. PLOS authors have the option to publish the peer review history of their article (what does this mean?). If published, this will include your full peer review and any attached files.

Reviewer #1: No

Reviewer #2: **Yes: **Mahmoud Shaaban, MD, PhD, FEACVI, FSCMR

Reviewer #3: No

---

## [Author Response · Author response to Decision Letter 0]

8 Feb 2024

Specific reviewer responses have been provided in an attached document. Additional edits from the editor have been addressed in the updated cover letter.

---

## [Decision Letter · Decision Letter 1]

5 Mar 2024

Enhanced echocardiographic assessment of intracardiac flow in congenital heart disease

PONE-D-23-30513R1

Dear Dr. Vlachos,

We’re pleased to inform you that your manuscript has been judged scientifically suitable for publication and will be formally accepted for publication once it meets all outstanding technical requirements.

Kind regards,

Hany Mahmoud Abo-Haded, MD

Academic Editor

PLOS ONE

**Comments to the Author**

1. If the authors have adequately addressed your comments raised in a previous round of review and you feel that this manuscript is now acceptable for publication, you may indicate that here to bypass the “Comments to the Author” section, enter your conflict of interest statement in the “Confidential to Editor” section, and submit your "Accept" recommendation.

Reviewer #1: All comments have been addressed

Reviewer #2: All comments have been addressed

2. Is the manuscript technically sound, and do the data support the conclusions?

Reviewer #1: Yes

Reviewer #2: (No Response)

3. Has the statistical analysis been performed appropriately and rigorously? 

Reviewer #1: Yes

Reviewer #2: (No Response)

4. Have the authors made all data underlying the findings in their manuscript fully available?

Reviewer #1: Yes

Reviewer #2: (No Response)

5. Is the manuscript presented in an intelligible fashion and written in standard English?

Reviewer #1: Yes

Reviewer #2: (No Response)

6. Review Comments to the Author

Reviewer #1: (No Response)

Reviewer #2: (No Response)

7. PLOS authors have the option to publish the peer review history of their article (what does this mean?). If published, this will include your full peer review and any attached files.

Reviewer #1: **Yes: **Jason Mandell

Reviewer #2: **Yes: **Mahmoud Shaaban, MD, PhD, FEACVI, FSCMR

---

## [Editor Report · Acceptance letter]

7 Mar 2024

PONE-D-23-30513R1 

PLOS ONE

Dear Dr. Vlachos, 

I'm pleased to inform you that your manuscript has been deemed suitable for publication in PLOS ONE. Congratulations! Your manuscript is now being handed over to our production team.

Kind regards, 

on behalf of

Professor Hany Mahmoud Abo-Haded 

Academic Editor

PLOS ONE